# Interplay Between Diet, Branched-Chain Amino Acids, and Myokines in Children: Vegetarian Versus Traditional Eating Habits

**DOI:** 10.3390/nu17050834

**Published:** 2025-02-27

**Authors:** Jadwiga Ambroszkiewicz, Magdalena Chełchowska, Joanna Mazur, Grażyna Rowicka, Witold Klemarczyk, Małgorzata Strucińska, Joanna Gajewska

**Affiliations:** 1Department of Screening Tests and Metabolic Diagnostics, Institute of Mother and Child, Kasprzaka 17a, 01-211 Warsaw, Poland; magdalena.chelchowska@imid.med.pl (M.C.); joanna.gajewska@imid.med.pl (J.G.); 2Department of Humanization in Medicine and Sexology, Collegium Medicum, University of Zielona Gora, 65-729 Zielona Gora, Poland; joanna.mazur@hbsc.org; 3Pediatric Gastroenterology Outpatient Clinic, Institute of Mother and Child, Kasprzaka 17a, 01-211 Warsaw, Poland; grazyna.rowicka@imid.med.pl; 4Department of Nutrition, Institute of Mother and Child, Kasprzaka 17a, 01-211 Warsaw, Poland; witold.klemarczyk@imid.med.pl (W.K.); malgorzata.strucinska@imid.med.pl (M.S.)

**Keywords:** branched-chain amino acids, lysine, myokines, FSTL-1, vegetarian diet, prepubertal children

## Abstract

**Background/Objectives:** The quality and composition of dietary proteins are crucial during growth, particularly in children who follow vegetarian diets. Branched-chain amino acids (BCAAs: leucine, isoleucine, and valine) and lysine play essential roles in muscle growth, repair, and metabolism and are involved in the regulation of muscle-derived proteins known as myokines. This study aimed to compare the dietary intake and circulating levels of BCAAs, lysine, and myokines—follistatin-like protein 1 (FSTL-1), myostatin, and myonectin—between vegetarian and omnivorous prepubertal children and to explore the impact of diet on muscle metabolism. **Methods:** Sixty-four healthy Caucasian children aged 4–9 years (forty-two vegetarians and twenty-two omnivores) were assessed for dietary intake using the Dieta 5^®^ (extended version Dieta 5.0) software. Circulating BCAAs and lysine were measured using high-performance liquid chromatography, while myokine concentrations were determined using enzyme-linked immunosorbent assays. **Results:** Vegetarian children showed significantly lower intakes of total protein, animal protein, BCAAs, and lysine than omnivores. Correspondingly, the circulating levels of isoleucine, valine, lysine, and albumin were significantly reduced in vegetarians. Among myokines, serum myostatin and myonectin levels were comparable between the groups, but vegetarians had significantly lower median FSTL-1 levels 7.7 (6.5–9.4) ng/mL than omnivores 9.7 (7.5–13.9) ng/mL (*p* = 0.012). In the entire group of children, positive correlations were observed between dietary total and animal protein intake and circulating valine and lysine levels. Dietary animal protein intake was also positively associated with the serum levels of all myokines, whereas plant protein intake was negatively correlated with myonectin concentration. **Conclusions:** In conclusion, vegetarian diets in prepubertal children are associated with reduced dietary protein quality and lower circulating BCAAs, lysine, and FSTL-1 levels, which may impact muscle metabolism. Optimizing vegetarian diets using high-quality plant proteins with proper essential amino acids could mitigate their deficiencies and support muscle development during critical growth periods.

## 1. Introduction

Protein and amino acids are fundamental to muscle health, playing a pivotal role in muscle physiology, including growth, repair, and metabolic activity [1,2]. Among the most extensively studied amino acids in this context are branched-chain amino acids (BCAAs), such as leucine, isoleucine, and valine. These essential amino acids are indispensable for protein synthesis, muscle recovery, and metabolic regulation [3]. Leucine, in particular, has been highlighted for its ability to stimulate muscle protein synthesis through the activation of the mammalian target of rapamycin (mTOR) signaling pathway, a key regulator of muscle mass. Isoleucine and valine complement leucine by supporting muscle function, metabolic stability, and energy homeostasis [4]. In addition to BCAAs, lysine, another essential amino acid, contributes significantly to muscle protein synthesis and recovery, especially when synergistically combined with other amino acids [5].

Protein and amino acids also regulate the secretion of myokine-signaling proteins released by muscle cells (myocytes) that mediate critical processes such as muscle function, metabolism, and homeostasis. Beyond their local effects on muscle tissue, myokines have systemic roles in regulating inflammation, insulin sensitivity, lipid metabolism, and overall metabolic health [6]. Factors such as physical activity and dietary patterns are known to modulate the secretion of myokines, making them central mediators of muscle health and systemic metabolism [7].

Several myokines are particularly relevant to muscle growth and function. Myostatin (MSTN), a well-characterized negative regulator of muscle growth, inhibits muscle development by activating pathways that promote protein degradation and suppress muscle cell proliferation [8,9,10]. Elevated myostatin levels can impair muscle hypertrophy and regeneration, highlighting its critical role in restricting excessive muscle growth [11,12,13]. Its regulatory function is particularly important during childhood, a period marked by rapid physical development and growth. Conversely, follistatin-like protein 1 (FSTL-1), a member of the tumor necrosis factor (TNF-β) superfamily, acts as a natural antagonist of myostatin. FSTL-1 facilitates muscle growth and hypertrophy by binding to and neutralizing myostatin. Its increased levels are associated with greater muscle mass and strength, underscoring its therapeutic potential in conditions involving muscle wasting or injury recovery [14].

Myonectin, also known as C1q/TNF-related protein isoform 5 (CTRPI5) or erythroferrone, is another crucial myokine primarily involved in energy regulation [15,16]. Unlike myostatin and FSTL-1, which are directly linked to muscle growth, myonectin is integral to glucose and lipid metabolism. Physical activity significantly influences myonectin secretion, suggesting its role in promoting energy homeostasis and preventing metabolic disorders such as obesity and type 2 diabetes [17]. In children undergoing dynamic growth, myonectin is particularly important for supporting healthy muscle function and metabolic balance.

The interplay between these myokines—myostatin, FSTL-1, and myonectin—highlights their collective importance in maintaining muscle mass, strength, and metabolic health [18,19]. Their secretion and activity are influenced by a range of factors, including physical activity, hormonal signaling, and dietary intake. Amino acids, particularly essential amino acids such as leucine, isoleucine, valine, and lysine, play an integral role in modulating myokine secretion and activity, ultimately influencing muscle health and systemic metabolism [3].

In recent years, the popularity of vegetarian diets has increased, especially in developed countries, with an increasing number of children raised in vegetarian households. Although this diet has various health benefits, not all researchers support its use, particularly in its more restrictive forms, such as veganism, for children [20,21,22,23,24]. Such dietary patterns may result in notable differences in amino acid profiles compared with traditional omnivorous diets [25,26,27]. However, in the current literature, there were few studies that specifically examined the relationship between vegetarian diets, amino acid intake, and myokine levels in children [28]. Our previous research revealed significant differences in dietary protein quality and serum amino acid profiles between children who adhere to vegetarian and omnivorous diets [29]. These findings highlight the need for further investigation of the impact of essential amino acids, particularly BCAAs and lysine, on muscle health and myokine regulation.

The present study aimed to assess the dietary intake, circulating concentrations of branched-chain amino acids and lysine, and serum levels of key myokines—FSTL-1, myostatin, and myonectin—in prepubertal children following vegetarian and traditional omnivorous diets. Additionally, it explored the potential relationships between serum myokines and essential amino acids, offering a better understanding of how dietary patterns may influence muscle metabolism during critical growth periods.

## 2. Materials and Methods

### 2.1. Subjects

This study, conducted from July 2022 to November 2024 at the Institute of Mother and Child in Warsaw, enrolled 64 healthy prepubertal children aged 4 to 9 years. The study group included 42 children (50% male) who had adhered to a vegetarian diet since birth. The children who had followed a vegetarian diet since birth came from families that adhered to a vegetarian lifestyle. Notably, 98% of the mothers and 72% of both parents were vegetarians. The vegetarian children regularly undergo medical and dietary check-ups at our institute. As their parents follow medical and nutritional guidelines, the children are healthy and developing well. Routine tests, including complete blood count, C-reactive protein (CRP), and ferritin levels, are conducted.

The control group consisted of 22 healthy children (50% male) following a traditional omnivorous diet that included meat, poultry, and fish.

In both groups, most of the examined children (approximately 80–90%) were from urban areas, came from families with a good economic status, and had parents with higher education degrees. Inclusion criteria required participants to follow a consistent vegetarian or omnivorous diet and be in the prepubertal stage. The exclusion criteria were low birth weight, developmental or nutritional disorders, gastrointestinal diseases, or regular use of medications, with the exception of standard vitamin D supplementation at a dose of 600–1000 IU/day (15–25 μg/day), as per updated guidelines for preventing and treating vitamin D deficiency in Poland [30]. Participants’ health status was evaluated through medical history, basic physical examination, and assessment of Tanner staging.

The study adhered to the ethical principles outlined in the Declaration of Helsinki and was approved by the Ethics Committee of the Institute of Mother and Child (approval number 15/2022, issued on 5 May 2022). Written informed consent was obtained from the parents of all the participants before enrollment in the study.

### 2.2. Anthropometric and Nutritional Measurements

Anthropometric data were collected using a calibrated stadiometer and electronic scale. Body weight and height were measured to calculate the body mass index (BMI). Nutritional intake was assessed using the Dieta5^®^ software (extended version Dieta 5.0) developed by the National Food and Nutrition Institute in Warsaw [31]. Parents were instructed by a nutritionist on how to maintain a food diary for their children. As detailed in a previous publication [32], from the ten-day record, three days with the most precise entries (two weekdays and one weekend day) with accurate entries were selected and entered into nutritional software to analyze the children’s average daily energy intake, as well as their macro and micronutrient consumption, including animal protein, plant protein, branched-chain amino acids, and lysine.

### 2.3. Biochemical Analyses

Venous blood samples were collected in the morning after an overnight fast to minimize diurnal variation. The samples were centrifuged at 2500× *g* for 10 min at 4 °C to separate the serum, which was then aliquoted and stored at −80 °C for up to two months prior to analysis. Biochemical parameters were measured in all children except for follistatin-1, which was analyzed in 95% of vegetarian and 91% of omnivorous subjects.

Amino acid concentrations were measured using high-performance liquid chromatography (HPLC) with reversed-phase separation and fluorescence detection (Shimadzu, Kyoto, Japan). The limit of detection for this method was 1 µmol/L, with coefficients of variation (CV) as follows: 1.55% for valine, 1.47% for leucine, 2.56% for isoleucine, and 6.61% for lysine. Amino acids were derivatized with o-phthalaldehyde (OPA) and 3-mercaptopropionic acid as primary amino acids and with 9-fluorenylmethyl chloroformate (FMOC) as the secondary amino acids. Fluorometric detection was performed using a Phenomenex Gemini C_18_ column (5 μm, 15 × 4.6 mm) (Torrance, CA, USA). Fluorescence excitation and emission wavelengths were set at 340 nm and 460 nm, respectively.

Serum myokine concentrations were quantified using commercially available human enzyme-linked immunosorbent assay (ELISA) kits according to the manufacturer’s protocols. FSTL-1 levels were measured using kits from Bioassay Technology Laboratory (Jiaxing, China) with a sensitivity of 0.025 ng/mL, intra-assay CV < 10%, and inter-assay CV < 12%. Myostatin was quantified using SunRed Biotechnology kits (Shanghai, China) with a detection limit of 0.05 ng/mL, and intra-assay and inter-assay precision were less than 8% and 11%, respectively. Myonectin levels were assessed using ELK Biotechnology kits (Wuhan, China) with a detection limit of 0.54 ng/mL and intra-assay and inter-assay CVs below 8% and 10%, respectively. Serum albumin levels were determined using ELISA kits from Assaypro LLC (St. Charles, MO, USA), with a detection limit of 0.27 mg/mL. The intra- and inter-assay coefficients of variation (CVs) were 4.9% and 10%, respectively. All samples were tested in duplicate.

### 2.4. Statistical Analyses

The distribution of variables was assessed for normality using the Kolmogorov-Smirnov test. Descriptive statistics are presented as mean ± standard deviation (SD) for variables with a normal distribution and as median with interquartile range (IQR) for variables that did not follow a normal distribution. Comparisons of biochemical parameters between the vegetarian and omnivorous groups were performed using the Mann–Whitney (MW) test. Given the small size of the control group, correlation analyses were conducted for the entire study population, as well as specifically within the vegetarian group, using Spearman’s rank correlation test.

The magnitude of differences between the groups was evaluated using the r effect size (ES), calculated as the absolute value of the standardized MW test (z) divided by the root of the total sample size (N). The effect sizes were interpreted using strict thresholds, with values of 0.30, 0.50, and 0.80 representing small, medium, and large effects, respectively.

In this type of study, sample size is determined by patient availability—in our case, prepubertal children who have followed a vegetarian diet since birth. We can only assess, in a statistical sense, the power of our study using a post hoc method. We conducted this analysis with the free G*Power (version 3.1.9.7) software, taking into account the Type I error rate, group sizes, and effect size. For a large effect size (0.8), the statistical power (1-β) was calculated to be 0.84905. This has implications for interpreting smaller effect sizes.

Differences were considered statistically significant at a *p*-value < 0.05. Statistical analyses were conducted using Statistical Package for the Social Sciences (SPSS) software (version 29.0, IBM Corp., Armonk, NY, USA).

## 3. Results

All participants were healthy, normal-weight, prepubertal, Caucasian children following vegetarian or omnivorous diets. Among the vegetarian children, 33 (79%) were lacto-ovo-vegetarians and nine (21%) were lacto-vegetarians. The study groups were comparable in age, sex, and anthropometric parameters (Table 1). Dietary analysis revealed that both groups exhibited similar total daily energy intake and percentage of energy from fat. However, vegetarian children had a significantly higher proportion of energy intake from carbohydrates (*p* < 0.01) and a significantly lower proportion of protein intake (*p* < 0.001) than omnivorous children. In addition, protein intake (in grams per day) and protein from animal sources were markedly lower (*p* < 0.001) in vegetarians than in meat-eaters. As expected, plant protein intake was significantly higher in vegetarians (*p* < 0.05). Additionally, vegetarian children had significantly reduced dietary intake of branched-chain amino acids and lysine compared with omnivores (*p* < 0.001). In vegetarians, the largest discrepancies were observed for lysine (48% lower), followed by isoleucine (39% lower), leucine (36% lower), and valine (24% lower).

Figure 1 shows the differences between the groups expressed as a positive effect size value, considering the seven parameters characterizing the diet from the lower part of Table 1. The largest differences were found for animal protein, followed by dietary lysine, total protein, and isoleucine intake, where ES significantly exceeded 0.5. Additionally, dietary valine intake showed a medium ES, and in the case of plant protein, the effect size did not exceed 0.3.

Biochemical analysis showed that vegetarian children had significantly lower serum levels of albumin (*p* < 0.001), as well as reduced concentrations of plasma isoleucine, valine, and lysine (*p* < 0.05) compared to omnivorous children (Table 2). In addition, leucine levels were slightly lower (but not significantly) in vegetarians than in omnivores. Serum myostatin and myonectin concentrations were similar across both dietary groups, whereas the level of FSTL-1 was significantly lower in vegetarian children.

Figure 2 shows a comparison of the data presented in Table 2 using the ES measure. The two biochemical parameters differed at levels > 0.3. For serum albumin and FSTL-1, the level of difference was qualified as large and medium, respectively. For the three parameters (lysine, isoleucine, and valine) with statistically significant differences between the two groups, ES was found to be less than 0.3.

Analyzing the correlations within the entire cohort of children, we observed that serum valine levels were significantly associated with dietary protein (*p* = 0.008), animal protein (*p* = 0.022), and lysine (*p* = 0.022) intake (Table 3). Furthermore, there were trends suggesting relationships between serum valine concentrations and the dietary intake of other BCAAs. Similarly, serum lysine levels significantly correlated with dietary energy intake (*p* = 0.022), total protein intake (*p* = 0.008), animal protein intake (*p* = 0.028), and leucine intake (*p* = 0.047). Additionally, serum lysine level was associated with dietary isoleucine and valine intake, but this did not reach statistical significance (*p* = 0.054 and *p* = 0.086, respectively).

In the studied children, serum albumin levels were positively correlated with the percentage of energy derived from protein (r = 0.419, *p* = 0.001) and animal protein intake (r = 0.335, *p* = 0.010) and negatively associated with plant protein intake (r = −0.332, *p* = 0.011).

Among the vegetarians, a significant positive correlation was observed between serum valine levels and protein intake (*p* = 0.003), as well as dietary intakes of leucine (*p* = 0.017), isoleucine (*p* = 0.020), valine (*p* = 0.014), and lysine (*p* = 0.032) (Table 4). Similarly, serum lysine levels were significantly correlated with energy, dietary protein, and BCAA intake (all *p* < 0.05). Although serum valine and lysine levels were also associated with both animal and plant protein intake, these relationships were not statistically significant. Notably, no significant correlations were identified for serum leucine or isoleucine levels in the entire group of children or vegetarians.

The relationships between serum myokine concentrations and nutritional or biochemical parameters in the entire group of children are detailed in Table 5. Among myokines, serum myostatin and myonectin levels showed a significant positive correlation (*p* = 0.038). Serum concentrations of all the studied myokines were significantly (*p* < 0.05) positively associated with dietary animal protein intake. Additionally, myonectin levels were negatively correlated with plant protein intake (*p* = 0.019). Serum FSTL−1 levels were not only positively correlated with animal protein intake but were also associated with dietary lysine intake (*p* = 0.045).

We did not observe significant associations between myokine concentrations and dietary or circulating essential amino acids in the group of vegetarian children.

## 4. Discussion

This study provides valuable insights into the impact of vegetarian and omnivorous diets on dietary intake, circulating essential amino acids, and myokine levels in prepubertal children. Our findings indicate that vegetarian children exhibit significantly lower amounts of total protein, animal protein, branched-chain amino acids, and lysine than their omnivorous peers, which is reflected in their reduced circulating levels. Notably, effect size analysis revealed that disparities in dietary intake were more pronounced than differences in serum amino acid concentrations, underscoring the dietary origin of these variations.

These results align with previous research, including our earlier study [29] and findings by other authors [7,25,27,33], who reported substantial differences in essential amino acid intake between plant-based and omnivorous diets. The reduced intake of lysine, a limiting amino acid in many plant proteins, is of particular concern. This deficiency has broader implications for skeletal muscle health, given the critical role of lysine in protein synthesis and collagen cross-linking [6]. Reduced availability of this amino acid may limit its role in muscle protein synthesis and metabolic regulation. Similarly, lower dietary intake of BCAAs, particularly leucine, may impair the activation of the mTOR pathway, a critical regulator of muscle protein synthesis that also inhibits protein degradation via the ubiquitin-proteasome system [6,34]. Konstantis et al. [35] observed that BCAAs had a beneficial effect on muscle mass and were associated with a significant increase in serum albumin concentration. In the present study, we found positive correlations between serum albumin levels and both total and animal protein intake, whereas plant protein intake was negatively correlated with albumin levels.

Isoleucine and valine, which contribute to protein metabolism, immune regulation, and energy production, were also consumed at lower levels by vegetarian children. This lower intake may adversely affect muscle growth and maintenance [7]. Our study showed that valine and lysine levels were positively correlated with dietary protein intake, specifically animal protein consumption, reinforcing the significance of protein sources in circulating amino acid levels. However, the correlations between these amino acid levels and plant protein intake did not reach statistical significance, suggesting lower bioavailability and suboptimal amino acid composition of plant-based proteins.

Serum albumin is a widely used biomarker for assessing nutritional status [36]. In a study of adult vegetarians, Caso et al. [37] observed about 13% lower albumin synthesis rate compared to individuals following an omnivorous diet, suggesting that different food sources may have varying effects on albumin synthesis. Albumin synthesis might be influenced by reduced amino acid availability, a consequence of the lower digestibility and amino acid score, as well as the higher fiber content of vegetarian diets. Similarly, previous research by Hovinen et al. [26] reported significantly lower serum albumin levels in vegan children.

In our study, vegetarian children, compared to omnivores, had significantly lower (approximately 18.5%) serum albumin levels but still in the normal range. Furthermore, albumin concentrations were positively correlated with the percentage of energy derived from protein (r = 0.419, *p* = 0.001) and animal protein intake (r = 0.335, *p* = 0.010) while negatively associated with plant protein intake (r = −0.3332, *p* = 0.011). The lower serum albumin levels observed in our vegetarian children might be related to the different proportions of animal/plant protein in their diets.

Myokines are pleiotropic factors involved in muscle cell proliferation, differentiation, mitochondrial function, inflammation, and metabolic homeostasis [18]. Consistent with our previous research [38,39], we found no significant differences in serum irisin, myonectin, fibroblast growth factor-21 (FGF-21), or myostatin levels between the vegetarian and omnivorous children. However, vegetarians exhibited higher levels of decorin—a myokine involved in collagen fibrillogenesis and inflammation regulation—suggesting a potential compensatory mechanism for maintaining muscle and connective tissue integrity.

In the present study, we found similar serum levels of myostatin and myonectin in both groups; however, we observed significantly lower FSTL-1 levels in vegetarian children. FSTL-1 is a critical antagonist of myostatin and activin that promotes muscle growth [14,40]. The reduced FSTL-1 levels observed in vegetarians may be linked to lower dietary protein quality and quantity, particularly lysine deficiency, which may influence its secretion. Given the essential role of lysine in collagen synthesis and muscle repair, this is particularly concerning for growing children, where muscle and bone development are critical. Lower leucine intake in vegetarian children may downregulate the mTOR pathway and potentially reduce FSTL-1 production. This aligns with studies demonstrating that essential amino acid supplementation, particularly lysine and leucine, improves muscle quality and reduces myostatin levels [6,40]. It can be excluded that the diet low in leucine observed in our vegetarians may downregulate this pathway, potentially reducing follistatin-1 production.

FSTL-1 is a protein with multidimensional actions that influence various biological processes [41]. It is synthesized in various organs and tissues, including adipose and muscle tissues, among others. By influencing myostatin, FSTL-1 plays a crucial role in muscle growth and maintenance, especially under conditions of energy deficiency or malnutrition. Its levels can be modulated by physical activity and nutritional status, highlighting its significance in muscle adaptation to environmental changes. This myokine exists in different isoforms (FST-288, FST-303, and FST-315), all of which have similar binding affinities to activins [19,42]. Interpretation of the levels of FSTL-1 is difficult due to its diverse actions (as both a myokine and adipokine), variability in measurement techniques, and the lack of established reference ranges in the pediatric population.

The observed alterations in amino acid and myokine profiles in vegetarian children may have several implications for skeletal muscle health. Reduced BCAA levels can impair muscle protein synthesis by modulating the mTOR pathway and suppressing the ubiquitin-proteasome system [33]. Furthermore, BCAAs reduce myostatin expression and inflammation via the SMAD and nuclear factor-kappa B pathways, further supporting their protective effects on muscle. Therefore, a diet low in BCAAs may disrupt the balance of muscle protein synthesis and degradation, indirectly affecting myokine release from muscle tissue.

It is well established that plant-based diets are associated with lower body mass index, reduced cholesterol levels, lower blood pressure, and a decreased risk of cardiovascular disease [43,44,45]. In our study, BMI did not differ significantly between the two groups and indicated normal nutritional status. Given the observed association between high protein intake in early childhood and an increased risk of obesity later in life, it cannot be ruled out that protein quality, in addition to quantity, also plays a role. This is particularly relevant to the high intake of animal protein, which has a greater impact on insulin growth factor-1 levels compared to plant protein [46,47]. However, confirming this hypothesis requires further well-designed prospective studies, as not all research supports this association.

Nutritional strategies that ensure a balanced diet, controlled calorie intake, and, if necessary, supplementation may offer promising approaches for maintaining muscle mass and preventing sarcopenia [48,49,50]. Studies in animal models have demonstrated that diet can influence the expression of key myokines, such as myostatin and myonectin, which regulate muscle signaling pathways [48]. Additionally, clinical research has shown the benefits of BCAAs and lysine supplementation in muscle preservation, reporting reduced myostatin levels and improved muscle function [6,51]. Further investigations are needed to clarify the precise mechanisms underlying these dietary effects and develop optimized nutritional guidelines for muscle health.

This study had several limitations. First, the cross-sectional design prevented us from establishing causal relationships between dietary patterns and myokine levels. Longitudinal studies are required to assess the long-term effects of vegetarian diets on muscle health. Second, the single-center study design and relatively small sample size limit the generalizability of our findings and constrain our ability to explore a wider range of variables or their interactions in detail. In future studies with a larger sample size, it would be valuable to estimate multivariate regression models for selected biochemical parameters that may have been influenced by diet type. This would allow us to determine whether the inclusion of a vegetarian diet remains a significant predictor of variability in a given parameter. Nonetheless, our recruitment of healthy prepubertal children adhering to a lacto-ovo- or lacto-vegetarian diet since birth provides a unique and well-defined study population. Third, our results relied on single measurements of myokines, which may not capture their long-term fluctuations. However, we employed robust laboratory techniques to ensure the high accuracy and reliability of the biochemical data. Importantly, this study is the first to report the serum FSTL-1 levels in vegetarian children, highlighting its novelty. Additional limitations include the absence of detailed body composition analyses (e.g., fat mass, lean mass, and bone mineral content) and reliance on self-reported physical activity data. However, we ensured that all participants were healthy, normal weight, and comparable in terms of physical activity. We are planning further studies on body composition assessments of adolescents following vegetarian diets to provide a more comprehensive understanding of musculoskeletal health. Despite these limitations, our study included both vegetarian and omnivorous children, thereby capturing meaningful dietary variability and enhancing the relevance of our findings.

## 5. Conclusions

In conclusion, vegetarian children exhibit lower dietary and circulating levels of essential amino acids such as lysine and BCAAs, which may impact muscle health and growth. Reduced FSTL-1 levels further underscore the importance of adequate protein intake and amino acid balance in vegetarian diets during childhood. Although plant-based diets offer numerous health benefits, optimizing protein quality and ensuring sufficient intake of essential amino acids are crucial. Future research should focus on the interplay between dietary patterns, amino acid profiles, and myokine status to develop tailored strategies to support muscle health.

## Figures and Tables

**Figure 1 nutrients-17-00834-f001:**
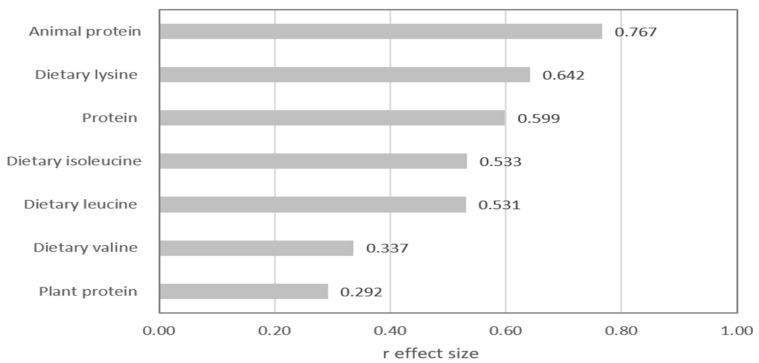
Mann–Whitney r effect size comparison of the two studied groups of children regarding nutritional parameters.

**Figure 2 nutrients-17-00834-f002:**
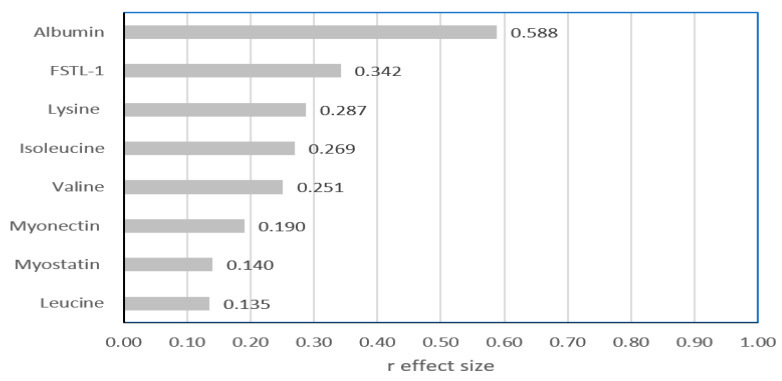
Mann–Whitney r effect size comparison of the two studied groups of children regarding biochemical parameters.

**Table 1 nutrients-17-00834-t001:** Anthropometric and dietary parameters of children following vegetarian and omnivorous diets.

	Vegetarians (n = 42)	Omnivores (n = 22)	*p* Value
Boys, n (%)	21 (50%)	11 (50%)	
Age (years)	6.3 (5.4–8.4)	6.0 (4.8–7.5)	0.197
BMI (kg/m^2^)	15.3 ± 1.4	15.2 ± 1.2	0.561
Energy (kcal/d)	1349 (1056–1634)	1476 (1261–1736)	0.210
Carbohydrates, % of energy	56.1 ± 4.5	52.0 ± 5.2	0.007
Fat, % of energy	29.2 ± 4.4	29.8 ± 5.6	0.867
Protein, % of energy	12.1 ± 2.2	15.9 ± 4.1	<0.001
Protein (g/d)	34.5 (27.8–42.3)	54.8 (44.8–67.7)	<0.001
Animal protein (g/d)	13.3 (5.2–17.6)	36.9 (27.5–46.7)	<0.001
Plant protein (g/d)	21.9 (18.1–26.3)	18.8 (14.7–22.2)	0.026
Dietary leucine (mg/d)	2734 (2063–3408)	4275 (3175–5338)	<0.001
Dietary isoleucine (mg/d)	1635 (1280–2083)	2670 (1924–3310)	<0.001
Dietary valine (mg/d)	2024 (1581–2584)	2670 (1924–3310)	0.010
Dietary lysine (mg/d)	1956 (1470–2376)	3762 (2755–4729)	<0.001

Data are reported as percentages (%); means ± standard deviations (SD) for normally distributed variables; medians and interquartile ranges (IQR) for skewed variables; BMI—body mass index.

**Table 2 nutrients-17-00834-t002:** Serum concentrations of biochemical parameters in children on vegetarian and omnivorous diets.

	Vegetarians	Omnivores	*p* Value
Albumin (mg/mL)	51.1 (45.7–56.5)	62.8 (56.6–68.2)	<0.001
Leucine (µmol/L)	109 (94–124)	115 (92–146)	0.280
Isoleucine (µmol/L)	51 (43–63)	57 (51–76)	0.031
Valine (µmol/L)	186 (166–209)	206 (176–267)	0.045
Lysine (µmol/L)	123 (88–151)	147 (115–174)	0.022
FSTL-1 (ng/mL)	7.7 (6.5–9.4)	9.7 (7.5–13.9)	0.012
MSTN (ng/mL)	1.1 (0.7–2.2)	1.4 (0.9–2.4)	0.264
Myonectin (ng/mL)	7.0 (4.8–8.7)	8.1 (6.4–8.9)	0.129

Data are reported as medians and interquartile ranges (IQR); FSTL-1—follistatin-like protein 1, MSTN—myostatin.

**Table 3 nutrients-17-00834-t003:** Spearman correlations between serum BCAAs and lysine concentrations and anthropometric and nutritional parameters in the entire group of studied children.

	Leucine		Isoleucine		Valine		Lysine	
	Rho *	*p*	Rho *	*p*	Rho *	*p*	Rho *	*p*
Age	−0.044	0.728	−0.024	0.849	−0.020	0.877	0.099	0.435
BMI	0.129	0.325	−0.023	0.860	0.078	0.551	0.042	0.753
Energy	0.094	0.475	−0.157	0.231	−0.109	0.407	−0.296	0.022
Protein	0.151	0.257	0.139	0.297	0.343	0.008	0.344	0.008
Animal protein	0.118	0.376	0.197	0.138	0.300	0.022	0.288	0.028
Plant protein	0.128	0.337	−0.039	0.773	0.132	0.322	0.111	0.408
Dietary leucine	0.037	0.780	0.059	0.658	0.251	0.057	0.262	0.047
Dietary isoleucine	0.035	0.796	0.062	0.643	0.256	0.052	0.254	0.054
Dietary valine	0.028	0.836	0.023	0.863	0.247	0.061	0.227	0.086
Dietary lysine	0.070	0.602	0.126	0.347	0.301	0.022	0.304	0.022

* Spearman’s rho; BMI—body mass index.

**Table 4 nutrients-17-00834-t004:** Associations between serum BCAAs and lysine concentrations and anthropometric and nutritional parameters in the vegetarian group.

	Leucine		Isoleucine		Valine		Lysine	
	Rho *	*p*	Rho *	*p*	Rho *	*p*	Rho *	*p*
Age	0.012	0.942	0.110	0.490	0.108	0.197	0.286	0.066
BMI	0.013	0.935	−0.113	0.476	−0.007	0.964	−0.081	0.610
Energy	−0.069	0.673	−0.186	0.251	−0.165	0.308	−0.377	0.018
Protein	0.304	0.064	0.184	0.268	0.467	0.003	0.461	0.004
Animal protein	0.201	0.226	0.157	0.345	0.309	0.059	0.289	0.078
Plant protein	0.232	0.161	0.059	0.727	0.294	0.073	0.321	0.057
Dietary leucine	0.224	0.176	0.143	0.393	0.385	0.017	0.409	0.011
Dietary isoleucine	0.214	0.196	0.147	0.377	0.375	0.020	0.398	0.013
Dietary valine	0.226	0.172	0.147	0.377	0.397	0.014	0.417	0.009
Dietary lysine	0.170	0.308	0.128	0.444	0.348	0.032	0.361	0.026

* Spearman’s rho; BMI—body mass index.

**Table 5 nutrients-17-00834-t005:** Spearman correlations between serum concentrations of myokines and anthropometric, nutritional, and biochemical parameters in the entire group of studied children.

	FSTL-1		MSTN		Myonectin	
	Rho *	*p*	Rho *	*p*	Rho *	*p*
Age	−0.008	0.957	−0.442	<0.001	−0.232	0.065
BMI	0.074	0.604	−0.069	0.601	0.076	0.565
Energy intake	−0.044	0.761	0.152	0.246	0.119	0.364
Protein intake	0.278	0.054	0.124	0.352	0.093	0.485
Animal protein	0.335	0.010	0.334	0.019	0.277	0.035
Plant protein	−0.141	0.334	−0.102	0.445	−0.306	0.019
Dietary leucine	0.253	0.080	0.141	0.292	−0.149	0.263
Dietary isoleucine	0.267	0.064	0.164	0.219	−0.128	0.339
Dietary valine	0.214	0.139	0.119	0.374	−0.165	0.216
Dietary lysine	0.288	0.045	0.229	0.084	0.049	0.714
Serum leucine	0.104	0.452	0.118	0.351	0.098	0.443
Serum isoleucine	0.140	0.313	0.045	0.722	0.146	0.251
Serum valine	0.155	0.264	0.101	0.429	0.107	0.402
Serum lysine	0.135	0.332	0.045	0.725	0.035	0.783
Serum albumin	0.215	0.119	0.188	0.136	0.177	0.162
FSTL-1	-	-	0.027	0.847	0.054	0.697
MSTN			-	-	0.261	0.038

* Spearman’s rho; BMI—body mass index; FSTL-1—follistatin-like protein 1, MSTN—myostatin.

## Data Availability

All data generated and analyzed in this study are included in this article. Further inquiries are available upon request from the corresponding author.

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
