# Peer review of "Interplay Between Diet, Branched-Chain Amino Acids, and Myokines in Children: Vegetarian Versus Traditional Eating Habits"

_nutrients, 2025, doi:10.3390/nu17050834_

Round 1
Reviewer 1 Report
Comments and Suggestions for Authors
The manuscript entitled “Interplay between diet, branched-chain amino acids and myokines in children: vegetarian versus traditional eating habits” aimed to compare the dietary intake and circulating levels of BCAAs, lysine, and myokines between vegetarian and omnivorous children and to investigate the impact of diet on muscle metabolism. Based on the obtained results, the authors concluded that vegetarian diets in children are associated with reduced dietary protein quality and lower circulating BCAAs, lysine, and follistatin-1 levels, which may impact muscle metabolism warranting better optimization of vegetarian especially during the development.
Can you please explain in more details the nature of the questioner used? Is it quantitative or semi-quantitative FFQ?
Were vegetarian children in the study lacto, ovo, or lacto-ovo vegetarians and in what ratio. Are there any differences if they use milk and dairy products?
Authors could give some recommendations on how to produce adequate protein intake in different vegetarian and/or vegan diets (e.g. from dairy or plant proteins such as legumes, quinoa and nuts).
Since the most straightforward effects of long-term vegetarian diet are a reduction in BMI, as well as reduction in cholesterol levels, and blood pressure is it possible that vegetarian and vegan diet could be beneficial in term of weight loss, especially in Europe where obesity and children obesity is becoming a problem.
Gajski G, Gerić M, Jakaša I, Peremin I, Domijan AM, Vučić Lovrenčić M, Kežić S, Bituh M, Moraes de Andrade V. Inflammatory, oxidative and DNA damage status in vegetarians: is the future of human diet green? Crit Rev Food Sci Nutr. 2023; 63(18): 3189-3221.
Technical remark:
Should Materials and methods section come as a the last based on the Manuscript template?
Author Response
Dear Reviewer,
Thank you very much for your valuable comments and suggestions, which have significantly contributed to improving our manuscript. Below, we provide detailed responses to each of your
Can you please explain in more details the nature of the questioner used? Is it quantitative or semi-quantitative FFQ?
We assessed dietary intakes using Diet Record method. Before medical visit, parents were asked to prepare a 10-day food diary for their children. They had been previously advised by the nutritionist on how to complete it accurately. The nutritionist then reviewed the diary to gather detailed information about the recorded foods and drink, including portion sizes and preparation methods, using a photo album of products and dishes. If necessary, corrections were made during the visit. From the 10-day record, three days with accurate entries (two weekdays and one weekend day) were selected. These data were then entered into the Dieta5® nutritional software (National Food and Nutrition Institute, Warsaw, Poland) to analyze the children`s average daily energy intake, as well as their macro- and micronutrient consumption.
We have added some information to the Materials and Methods section, while a detailed description of the method was provided in our previous study: Ambroszkiewicz et al. Biol. Trace Elem. Res. 2017, 180, 182–190.
Were vegetarian children in the study lacto, ovo, or lacto-ovo vegetarians and in what ratio. Are there any differences if they use milk and dairy products?
Among the vegetarian children included in our study, 33 participants (79%) were lacto-ovo-vegetarians, while 9 participants (21%) were lacto-vegetarians. Importantly, all vegetarian participants consumed milk and dairy products. This information has been stated in the first sentence of the Results section.
Authors could give some recommendations on how to produce adequate protein intake in different vegetarian and/or vegan diets (e.g. from dairy or plant proteins such as legumes, quinoa and nuts).
We understand the importance of providing dietary recommendations to ensure adequate protein intake for children on vegetarian or vegan diets, supporting their overall growth and development. To optimize protein intake, we recommend incorporating a variety of plant-based protein sources. For vegetarian children who consume dairy and eggs, these foods further enhance protein quality. Some of the best plant-based sources include legumes (such as beans, soy, peas, and chickpeas), whole grains like buckwheat and bulgur, and nuts like almonds.
Since the most straightforward effects of long-term vegetarian diet are a reduction in BMI, as well as reduction in cholesterol levels, and blood pressure is it possible that vegetarian and vegan diet could be beneficial in term of weight loss, especially in Europe where obesity and children obesity is becoming a problem.
Gajski G, Gerić M, Jakaša I, Peremin I, Domijan AM, Vučić Lovrenčić M, Kežić S, Bituh M, Moraes de Andrade V. Inflammatory, oxidative and DNA damage status in vegetarians: is the future of human diet green? Crit Rev Food Sci Nutr. 2023; 63(18): 3189-3221.
We greatly appreciate this insightful comment regarding the potential benefits of vegetarian diets. It is well-established that plant-based diets are associated with lower BMI, reduced cholesterol levels, and lower blood pressure, as well as a reduced risk of cardiovascular diseases. In our study, the BMI of children in both groups did not differ significantly and indicated a normal nutritional status. Considering observations regarding the association between high protein intake in early childhood and the later development of obesity, it cannot be ruled out that, in addition to the quantity, the quality of protein also plays a role. This appears to be particularly relevant to the high intake of animal protein. This is explained by the greater effect of animal protein compared to plant protein on IGF-1 levels. To confirm this hypothesis, further well-designed prospective studies are necessary, as not all observations support this association.
We have expanded the Discussion section to include this perspective, citing relevant literature to support the argument [Günther et al. Am J Clin Nutr 2007; 86(6): 1765-1772; Kahleova et al. Nutrients 2017, 9, 848; Gajski et al. Crit Rev Food Sci Nutr. 2023; 63(18): 3189-3221; Thorisdottir et al. Acta Paediatr 2014; 103: 512-517; Pimpin et al. Br J Nutr 2018; 120(7): 820-829].
Technical remark:
Should Materials and methods section come as a the last based on the Manuscript template?
In response to your technical remark, we confirm that, according to the guidelines of Nutrients, the Materials and Methods section should follow the Introduction section. We acknowledge that different journals follow different conventions—for instance, in International Journal of Molecular Sciences, the Materials and Methods section is placed at the end. However, we have ensured that our manuscript complies with the formatting requirements of Nutrients.
Once again, we sincerely appreciate your constructive feedback, which has allowed us to refine our manuscript.
Best regards,
Jadwiga Ambroszkiewicz
Department of Screening Tests and Metabolic Diagnostics
Institute of Mother and Child, email: jadwiga.ambroszkiewicz@imid.med.pl
Reviewer 2 Report
Comments and Suggestions for Authors
The manuscript by Ambroszkiewicz et al on “Interplay between Diet, Branched-Chain Amino Acids and Myokines in Children: Vegetarian Versus Traditional Eating Habits” is well written and describes interesting associations between dietary protein intake and serum amino acids and myokines. Thanks for the opportunity to review.
Nevertheless, there are some points I would like to suggest for consideration
Sample size: the study is focussed on the detection of associations and group difference, for this type of study it seems important to have a power calculation before data collection, this seems not to be the case, so should be discussed
Statistics: Mann-Whitney test and Spearman correlation were used, but the data analysis might benefit from using more complex models considering adjustment for potential confounders or interaction effects. If total protein intake is identified as a relevant factor, would one adjust for this, when looking for the effect of a specific amino acid?
Maybe state that the study is explorative and so the point multiple testing is not considered.
Group differences: omnivorous and vegetarian groups often differ also in other factors (e.g. family income, parental education) besides diet, so would be good to show more details of the groups
The clearest difference between the study groups seems to be albumin level, but this is not discussed at all.
Measurements of irisin and FGF21 are mentioned, but data not shown. These are widely applied markers and data seem relevant, independent of the significance of a group difference. Does this indicate that albumin responds more sensitive to protein intake than FGF21?
Specific points
Lines 59-61: please give a reference
Line 114: it seems good to describe already here that children were on vegetarian diet since birth, do also parents follow vegetarian diet?
Line 119: any laboratory data indicating health status, such as blood count or CRP, ferritin?
Line 131: dietary recall or dietary protocols? If diet on 10 days was recorded, why only 3days evaluated? How were these days selected?
Line 142: would be good to give here similar to the ELISAs information about analytical precision
Table 1: better <0.001 than 0.000
Tables 3, 4, 5: please indicate which of the values are significant, e.g. bold font
Line 290: is it deficiency or lower intake?
Author Response
Dear Reviewer,
We sincerely appreciate your thoughtful comments and valuable suggestions, which have significantly contributed to improving our manuscript. Below, we provide detailed responses to each of your points.
Nevertheless, there are some points I would like to suggest for consideration
Sample size: the study is focused on the detection of associations and group difference, for this type of study it seems important to have a power calculation before data collection, this seems not to be the case, so should be discussed.
In this type of study, the sample size is determined by the availability of participants—in our case, prepubertal children who had followed a vegetarian diet since birth. The statistical power of our study can only be assessed post hoc. We conducted such an analysis using G*Power software, taking into account the Type I error rate, group sizes, and effect size. For a large effect size (0.8), the statistical power (1-β) was 0.84905. This has implications for interpreting smaller effect sizes. Corrections have been made in the Materials and Methods section accordingly.
Statistics: Mann-Whitney test and Spearman correlation were used, but the data analysis might benefit from using more complex models considering adjustment for potential confounders or interaction effects. If total protein intake is identified as a relevant factor, would one adjust for this, when looking for the effect of a specific amino acid?
Maybe state that the study is explorative and so the point multiple testing is not considered.
We took into account the entire panel of nutritional and biochemical parameters. Therefore, there is no primary outcome measure that would serve as the dependent variable in the multivariate analysis.
Group differences: omnivorous and vegetarian groups often differ also in other factors (e.g. family income, parental education) besides diet, so would be good to show more details of the groups
In both groups, most of the examined children (approximately 80–90%) were from urban areas, came from families with a good economic status, and had parents with higher education degrees.
We added this information in Material and Methods section.
The clearest difference between the study groups seems to be albumin level, but this is not discussed at all.
Serum albumin is a widely used biomarker for assessing nutritional status. In a study of adult vegetarians, Caso et al. reported an approximately 13% lower albumin synthesis rate compared to individuals following an omnivorous diet. Similarly, previous research by Hovinen et al. found significantly lower serum albumin levels in vegan children. In our study, vegetarian children had serum albumin levels approximately 18.5% lower than those of omnivorous children.
Moreover, serum albumin levels were positively correlated with the percentage of energy derived from protein (r = 0.419, p = 0.001) and animal protein intake (r = 0.335, p = 0.010), while they were negatively associated with plant protein intake (r = -0.332, p = 0.011).
We have expanded the Discussion section to include these findings.
Measurements of irisin and FGF21 are mentioned, but data not shown. These are widely applied markers and data seem relevant, independent of the significance of a group difference. Does this indicate that albumin responds more sensitive to protein intake than FGF21?
You correctly noted that we mentioned irisin and FGF-21 without presenting the corresponding data. These markers were referenced based on our previous findings, published in J Clin Med (2021) and Nutrients (2024), where we observed no statistically significant differences in their serum levels between vegetarian and omnivorous participants.
FGF-21 plays a key role in glucose and lipid metabolism and is responsive to nutrient intake. Its secretion is stimulated by feeding and suppressed by fasting, while physical activity also influences its levels. It is not excluded that albumin levels are more sensitive to protein intake than FGF-21, but other factors—particularly physical activity—may influence FGF-21 levels and introduce compensatory effects.
However, as irisin and FGF-21 were not the primary focus of this study, we did not present the data in detail.
Specific points:
Lines 59-61: please give a reference
A reference has been added both in the text and in the reference list.
Line 114: it seems good to describe already here that children were on vegetarian diet since birth, do also parents follow vegetarian diet?
We have clarified that the children who had followed a vegetarian diet since birth came from families that adhered to a vegetarian lifestyle. Notably, 98% of the mothers and 72% of both parents were vegetarians.
This information has been added to the Materials and Methods section.
Line 119: any laboratory data indicating health status, such as blood count or CRP, ferritin?
The vegetarian children in our study regularly undergo medical and dietary check-ups at our institute. As their parents follow medical and nutritional guidelines, the children are healthy and developing well. Routine tests, including complete blood count, C-reactive protein (CRP), and ferritin levels, are conducted. The results of these tests in vegetarian children were previously published in: Ambroszkiewicz et al. Biol. Trace Elem. Res. 2017, 180, 182–190.
Line 131: dietary recall or dietary protocols? If diet on 10 days was recorded, why only 3 days evaluated? How were these days selected?
We assessed dietary intakes using Diet Record method. Before medical visit, parents were asked to prepare a 10-day food diary for their children. They had been previously advised by the nutritionist on how to complete it accurately. The nutritionist then reviewed the diary to gather detailed information about the recorded foods and drink, including portion sizes and preparation methods, using a photo album of products and dishes. If necessary, adjustments were made to ensure accuracy during the visit. From the 10-day record, three days with the most precise entries (two weekdays and one weekend day) were selected. These data were then entered into the Dieta5® nutritional software (National Food and Nutrition Institute, Warsaw, Poland) to analyze the children`s average daily energy intake, as well as their macro- and micronutrient consumption.
A detailed description of the method was provided in our previous study Ambroszkiewicz et al. Biol. Trace Elem. Res. 2017, 180, 182–190.
Line 142: would be good to give here similar to the ELISAs information about analytical precision
We have now included analytical precision details in the Materials and Methods section.
Table 1: better <0.001 than 0.000
We have changed 0.000 to <0.001, following your suggestion
Tables 3, 4, 5: please indicate which of the values are significant, e.g. bold font
We have now highlighted significant values in bold to improve clarity.
Line 290: is it deficiency or lower intake?
Thank you for this correction—we have changed "deficiency" to "lower intake" to reflect the accurate interpretation
Once again, we sincerely appreciate your constructive feedback, which has helped us refine and strengthen our manuscript.
Best regards,
Jadwiga Ambroszkiewicz
Department of Screening Tests and Metabolic Diagnostics
Institute of Mother and Child, email: Jadwiga.ambroszkiewicz@imid.med.pl
Round 2
Reviewer 2 Report
Comments and Suggestions for Authors
Thank you very much for discussing/considering the points made
Author Response
Thank you very much.